# Role of Dietary Defatted Rice Bran in the Modulation of Gut Microbiota in AOM/DSS-Induced Colitis-Associated Colorectal Cancer Rat Model

**DOI:** 10.3390/nu15061528

**Published:** 2023-03-22

**Authors:** Laleewan Tajasuwan, Aikkarach Kettawan, Thanaporn Rungruang, Kansuda Wunjuntuk, Pinidphon Prombutara

**Affiliations:** 1Graduate Student in Doctor of Philosophy Program in Nutrition, Faculty of Medicine Ramathibodi Hospital and Institute of Nutrition, Mahidol University, Bangkok 10400, Thailand; laleewan.taa@student.mahidol.ac.th; 2Institute of Nutrition, Mahidol University, Nakhon Pathom 73170, Thailand; 3Department of Anatomy, Faculty of Medicine Siriraj Hospital, Mahidol University, Bangkok 10700, Thailand; thanaporn.run@mahidol.ac.th; 4Department of Home Economics, Faculty of Agriculture, Kasetsart University, Bangkok 10900, Thailand; fagrkdw@ku.ac.th; 5OMICS Sciences and Bioinformatics Center, Faculty of Science, Chulalongkorn University, Bangkok 10330, Thailand; pinidphon.p@chula.ac.th

**Keywords:** defatted rice bran, colitis-associated colorectal cancer, gut microbiota, fiber-degrading bacteria, short-chain fatty acid (SCFA)-producing bacteria, short-chain fatty acids (SCFAs), goblet cell loss, mucus layer thickness, health, health product

## Abstract

Defatted rice bran (DRB) is a by-product of rice bran derived after the oil extraction. DRB contains several bioactive compounds, including dietary fiber and phytochemicals. The supplementation with DRB manifests chemopreventive effects in terms of anti-chronic inflammation, anti-cell proliferation, and anti-tumorigenesis in the azoxymethane (AOM) and dextran sodium sulfate (DSS)-induced colitis-associated colorectal cancer (CRC) model in rats. However, little is known about its effect on gut microbiota. Herein, we investigated the effect of DRB on gut microbiota and short chain fatty acid (SCFA) production, colonic goblet cell loss, and mucus layer thickness in the AOM/DSS-induced colitis-associated CRC rat model. The results suggested that DRB enhanced the production of beneficial bacteria (*Alloprevotella*, *Prevotellaceae UCG-001*, *Ruminococcus*, *Roseburia*, *Butyricicoccus*) and lessened the production of harmful bacteria (*Turicibacter*, *Clostridium sensu stricto 1*, *Escherichia–Shigella*, *Citrobacter*) present in colonic feces, mucosa, and tumors. In addition, DRB also assisted the cecal SCFAs (acetate, propionate, butyrate) production. Furthermore, DRB restored goblet cell loss and improved the thickness of the mucus layer in colonic tissue. These findings suggested that DRB could be used as a prebiotic supplement to modulate gut microbiota dysbiosis, which decreases the risks of CRC, therefore encouraging further research on the utilization of DRB in various nutritional health products to promote the health-beneficial bacteria in the colon.

## 1. Introduction

Colorectal cancer (CRC) remains an unresolved global public health burden, ranking third in incidence and mortality worldwide. According to a recent study, CRC cases will reach around 3.2 million in 2040 [1]. Genetics, environmental factors, and gut microbiota have been related to CRC development and progression. CRC occurs in the colon; therefore, it is closely linked to gut microbiota changes. The gut microbiota is vital in maintaining intestinal homeostasis [2]. While gut microbiota dysbiosis (changes in microbial communities) disrupts intestinal homeostasis, leading to increased intestinal epithelium permeability, bacterial pathogens invade the epithelial cells quickly and stimulate the host immune responses [3,4,5]. The binding between microbial ligands (known as microbial associated molecular patterns (MAMPs)) such as lipopolysaccharide (LPS) and peptidoglycan (PGN), present on the bacterial cell surface, and pattern recognition receptors (PRRs) such as toll-like receptors (TLRs) and Nod-like intracellular receptors (NODLRs), present on the host immune cells, results in stimulating host immune responses and producing pro-inflammatory cytokines, which protects the intestinal epithelial cells from bacterial invasion [6]. The prolonged production of pro-inflammatory cytokines induces chronic inflammation in the colon, which can develop to CRC. Therefore, maintaining a healthy gut microbiota with prebiotic or probiotic supplements for the excellent health of colon cells has emerged as an alternative treatment for CRC.

Rice bran is the outer layer of milled rice, produced as a by-product in the rice milling process. It contains many nutrients, including starch, dietary fibers, lipids, proteins, vitamins, and minerals [7,8]. Nowadays, rice bran has been widely employed in the manufacturing of rice bran oil. During the industrial production of rice bran oil, large amounts of DRB are produced. During the oil extraction process, some essential nutrients are removed, but DRB still retains compounds of high nutritional value, such as soluble and insoluble dietary fiber, and some phytochemicals [9]. In particular, dietary fiber, an essential component of DRB, has been reported to be associated with increased intestinal microbiota community and could be protected against CRC [10]. In an animal study, Huan Wang et al. determined that finishing pigs fed with 7% of DRB as a substitute for corn had a beneficial effect on the thickness of intestinal wall, and increased *Bifidobacterium* and *Clostridium perfringens* in the colon [11]. In a similar study, 10% fermented DRB supplementation in finishing pigs for 30 days enhanced the gut microbial richness and the abundance of fiber-degrading bacteria (*Clostridium butyricum* and *Lactobacillus amylovorus*). Moreover, supplementation of 10% fermented DRB also significantly elevated the short-chain fatty acids (acetate and butyrate) in feces [12]. More recently, the insoluble dietary fiber, extracted from DRB by enzymatic treatments, could restore the reduction in species of gut microbiota caused by high-fat diet, increase the richness of the microbial community, and alter the metabolic function of gut microbiota on hyperlipidemia in rats fed with a high-fat diet [13].

Dietary fiber fermented by bacterial enzymes is converted into short-chain fatty acids (SCFAs), primarily acetate, propionate, and butyrate, representing the 90–95% of the SCFAs present in the colon. The fiber has also shown prebiotic properties in several studies. High dietary fiber intake increases butyrate-producing bacteria and SCFAs production in the colon. SCFAs have been suggested as the key metabolites linking the gut microbiota to inflammation and CRC. SCFAs are absorbed across the epithelial cells (colonocytes) through passive diffusion, and active transport mediated by monocarboxylate transporters 1 (MCT1) and sodium-coupled monocarboxylate transporter 1 (SMCT1) to maintain intestinal homeostasis and generate adenosine triphosphate (ATP) for colonocytes [14]. In addition, SCFAs act as endogenous ligands for G protein-coupled receptors (GPCRs), and intracellular SCFAs affect gene expression by inhibiting the histone deacetylase (HDAC) [15]. The study of SCFAs, especially propionate and butyrate, has highlighted their effects on immune modulation and inflammatory responses. For instance, propionate and butyrate regulate the T cell function through G-protein-coupled receptors (GPR43 and GPR109A), and by inhibition of histone deacetylase (HDAC), which hampers the activation of the NF-κB signaling pathway [16,17]. Likewise, these SCFAs also inhibit the pro-inflammatory cytokine production (IL-6, IL-8, IL-1β, and TNF-α) by leucocytes [16,18]. Moreover, propionate and butyrate affect the differentiation of regulatory T cells (FOXP3) and the production of IL-10, which reduce inflammation [16]. SCFAs also exert anti-carcinogenesis effects in the colon by promoting apoptosis and suppressing the proliferation of tumor cells through the Wnt/β-catenin signaling pathway by inhibiting HDAC activity [14,19,20]. Numerous studies have shown that butyrate is critical for modulating immune and inflammatory responses and mucus barrier function [21]. Moreover, butyrate stimulates mucin secretion, which is essential for mucoprotection.

Therefore, the present study investigates the effect of dietary DRB on gut microbiota and SCFAs production, colonic goblet cell loss, and mucus layer thickness in the AOM/DSS-induced colitis-associated CRC model in rats.

## 2. Materials and Methods

### 2.1. Defatted Rice Bran (DRB)

DRB was obtained from Thai Ruam Jai Vegetable Oil Co., Ltd. (Phra Nakhon Si Ayutthaya, Thailand). It was procured from a mix of brown Thai rice varieties in the central area of Thailand. After rice milling, the bran was extracted with hexane. DRB was powdered and heated to lower moisture after the oil extraction process, then passed through a 60-mesh sieve and kept in a sealed container under a hygienic condition at −20 °C until further use.

### 2.2. Animal Experiment and Sample Collection

All animal experiments followed the ethical procedure guidelines from Siriraj Animal Care and Use Committee, Mahidol University (COA no. 004/2562). Wistar male rats (age: four weeks old, weight: 90–100 g) were obtained from Nomura Siam International Co., Ltd. (Bangkok, Thailand). All rats were housed in individually ventilated plastic cages (2 rats/cage) under standard conditions (temperature 23 ± 1 °C; humidity 50 ± 10% and 12 h-light/dark cycle) and were allowed access to a standard commercial diet and water.

Experimental animal design of the study is shown in Figure 1A. After a one-week adaptation period, rats were randomly administered into six groups: (1) control, (2) defatted rice bran 3 g (DRBL), (3) defatted rice bran 6 g (DRBH), (4) induction, (5) induction + DRBL, and (6) induction + DRBH. The rats of groups 2, 3, 5, and 6 were administered gavaged feeding with 3 and 6 g/kg DRB daily throughout the study, while the control group and induction group received orally sterile water daily. The dose of DRB followed the previous studies [22,23] and then converted to an animal equivalent dose using the ratios of human and rat body surface area [24].

All rat’s body weight and food intake were daily recorded throughout the study (Appendix A). After two weeks, the rats of groups 4, 5, and 6 were subcutaneously injected with 15 mg/kg AOM (Sigma-Aldrich Pte. Ltd., Singapore) once weekly for 2 weeks. One week later, the rats received 4% (*w*/*v*) DSS (TdB Consultancy, Uppsala, Sweden, molecular weight 36–50 kDa) in drinking water for one week, followed further by one week of recovery with regular water and repeated one time for the DSS induction period.

All rats were euthanized by CO_2_ asphyxiation at ten weeks after the first AOM injection. The cecal and colonic luminal contents were collected and stored at −20 °C until further use. In this study, the colonic luminal contents were used and represented as a fecal sample. Afterward, the colon specimens were gently washed with cold phosphate-buffered saline solution (PBS) and the colon’s weight and length were measured (Appendix A). Then, the surface of the colonic mucosa was scraped vigorously with a sterile scalpel blade, and the tumor tissues (Figure 1B) were removed and stored at −20 °C until further use for microbiome analysis. The remaining colon specimens were kept in 10% neutral phosphate-buffered formalin for alcian blue/periodic acid-Schiff staining.

### 2.3. Colonic Goblet Cells and Mucus Layer Thickness Evaluation

Post 10% neutral phosphate-buffered formalin fixations (at least 24 h), colon specimens were dehydrated in a series of graded ethanol and cleared with xylene. Then, the colons were embedded in a paraffin block, and the transversal sections were cut into 4 µm using a microtome. The paraffin-embedded sections were placed on glass slides, then de-paraffinized with xylene and rehydrated with absolute and 95% ethanol. The colonic sections were stained with alcian blue solution (pH 2.5) for 30 min, periodic acid for 10 min and Schiff solution for 10 min, respectively. The colonic goblet cell and mucus layer thickness were observed under an Olympus SC 180 microscope (Olympus, Shinjuku-ku, Tokyo, Japan) using 4×, 10×, and 40× objectives, and images were captured. Ten different fields per section and three sections per rat were randomly chosen for evaluation. The goblet cell loss is defined as reducing goblet cell numbers relative to baseline goblet cell numbers per crypt. The goblet cell loss was evaluated by counting the number of goblet cells per crypt. At the same time, the mucus layer thickness was measured using Image J software version 1.52a (National Institutes of Health, Bethesda, MD, USA), with the observer being blinded to the analysis.

### 2.4. Gut Microbiota Analysis

#### 2.4.1. DNA Extraction

Metagenomic DNA in colonic luminal contents (feces), mucosa surfaces, and tumor tissues were extracted using the QIAamp Power Fecal Pro DNA kit (Qiagen, Germantown, MD, USA), according to the manufacturer’s instructions. Before DNA extraction, the tumor tissues were lysed using the MasterPure Complete DNA and RNA Purification kit (Lucigen, Middleton, WI, USA), following the manufacturer’s protocol. Negative controls were always performed during extraction to verify this process without contamination. Purification and concentration of the DNA quantified using a NanoDrop spectrophotometer (Implen GmbH, München, Bayern, Germany), and DNA quality was checked by 1% agarose gel electrophoresis. The DNA samples were stored at −20 °C until further analysis.

#### 2.4.2. 16S rRNA Amplicon Library Preparation and Sequencing

The 16S rRNA gene was amplified from metagenomic DNA samples using a primer targeting the V3–V4 region (16S Forward Primer 5′-TCGTCGGCAGCGTCAGATGTGTATAAGAGACAGCCTACGGGNGGCWGCAG-3′ and 16S Reverse Primer 5′-GTCTCG TGGGCTCGGAGATGTGTATAAGAGACAGGACTACHVGGTATCTAATCC-3′) (Macrogen Inc., Gangnam-gu, Seoul, Republic of Korea) by a thermocycler PCR system (PCRmax Alpha Cycler, Staffordshire, UK).

The PCR reaction for colonic luminal contents was composed of a DNA template (2 µL), primers (0.5 µL), ultrapure distilled water (9.5 µL) (Invitrogen, Waltham, MA, USA), and 2× HiFi PCR master mix (12.5 µL) (sparQ HiFi PCR master mix, Quantabio, Beverly, MA, USA), while for mucosa and tumor tissue, the PCR reaction was composed of a DNA template (5 µL), primers (0.5 µL), ultrapure distilled water (6.5 µL), and 2× HiFi PCR master mix (12.5 µL). Negative controls were always performed to verify the process without contamination. PCR was performed under the following conditions: an initial denaturing step at 98 °C for 2 min followed by 28 amplification cycles (for luminal content samples) and 30 amplification cycles (for mucosa and tumor tissue samples) at 98 °C for 20 s, 60 °C for 30 s, and 72 °C for 1 min, and a final extension step at 72 °C for 1 min. 16S V3 and V4 amplicons were verified by agarose gel electrophoresis. Afterwards, PCR products were purified using the sparQ PureMag beads (Quantabio, Beverly, MA, USA). The purified 16S amplicon was indexed using the 2× HiFi PCR master mix and 2.5 µL of each Nextera XT index primer (Nextera XT Index Kit v2, Illumina Inc., San Diego, CA, USA) in a 25 µL PCR reaction, followed by 8 cycles of PCR condition. 16S index amplicons were verified by agarose gel electrophoresis. Subsequently, the purified 16S index library was pooled at a concentration of 4 nM, measuring concentration by Qubit ds DNA HS assay kits (Invitrogen, Eugene, OR, USA) and diluted to final loading concentration at 5 pM, using pre-chilled hybridization buffer (HT1) (Illumina Inc., San Diego, CA, USA). Finally, 25% of the whole-genome sequencing (WGS) control library was spiked into the pooled library and added to the Miseq Reagent Kit v2 (500 cycles) (Illumina Inc., San Diego, CA, USA). Cluster generation and 250 bp paired-end read sequencing were performed on an Illumina MiSeq platform at Omics Sciences and Bioinformatics Center (Chulalongkorn University, Bangkok, Thailand).

#### 2.4.3. Bioinformatics Analysis

FATSQ raw data were generated and de-multiplexed using Miseq reporter software version 3.1. Targeted V3–V4 primer sequences were removed or trimmed, and the data were imported to Quantitative Insights Into Microbial Ecology 2 (QIIME2) software version 2019.7. The imported sequencing reads were preprocessed using the DADA2 program. Denoised reads were clustered into amplicon sequence variants (equivalent to OTUs), and the sequence reads were assigned to operational taxonomic units (OTUs) at 97% nucleotide identity. Then, a phylogenetic tree was built using SEPP QIIME 2 plugin. A rarefaction curve was created, and the Pielou’s evenness, Faith’s phylogenetic diversity, observed OTUs, and Shannon diversity index were also measured alpha diversity within microbial communities. Beta diversity metrics, including Bray–Curtis dissimilarity, Jaccard index, and weighted and unweighted UniFac distance were calculated microbial communities in each sample, and principal coordinate’s analysis (PCoA) plots were generated. Finally, taxonomy was assigned to the OTUs using a Naive–Bayes approach implemented in the scikit learn Python library and the SILVA database, and classification stacked bar plots were created.

### 2.5. Short-Chain Fatty Acid (SCFA) Analysis

The cecal SCFA concentrations, including acetic, propionic, and butyric acid, were measured according to a modified method by Ribeiro W.R. et al. [25]. The cecal SCFAs were extracted using a liquid–liquid extraction procedure. Approximately 20 mg of cecal contents was removed to microtubes and placed on ice. A total of 200 µL of distilled water was added to the cecal contents and then homogenized using a metal spatula. A total of 20 mg of citric acid, 40 mg of NaCl, 40 µL of 0.1 M HCl, and 200 µL of organic solvents (N-butanol, tetrahydrofuran, and acetonitrile) in 5:3:2 ratios were added into the homogenate samples and mixed vigorously using the vortex for 1 min. The mixture samples were centrifuged at 15,000× *g* at 4 °C for 10 min. The supernatant was transferred to chromatographic vials equipped with 200 μL inserts and stored at −20 °C until further analysis.

The supernatant (2 μL) was injected in a 50:1 split ratio into a gas chromatograph (model 6890N; Agilent Technologies, Santa Clara, CA, USA) equipped with a flame ionization detector (FID) and a capillary column (DB-23, 60 m × 0.25 mm × 0.25 µm) coated with a film of 0.25 µm composed of 74.5% 1-methyl-naphthalene. Helium was used as a carrier gas. The initial oven temperature was maintained at 70 °C for 5 min and increased at 5 °C/min to 80 °C for 5 min. The injector and detector temperatures were set at 250 °C. The SCFA contents in cecal were identified by retention time compared to SCFAs standard (acetic (71251), propionic (94425), and butyric acid (19215), Sigma-Aldrich Pte. Ltd., Singapore). The EZChrom software version 3.3.1 was used to integrate the peak areas. The SCFA concentrations (mg/mL) in each sample were determined by comparing their peak areas with standard calibration curves. Finally, the mg/mL unit was converted to mmol/kg. This procedure calculated a percent recovery range of 90–110 for quality control.

### 2.6. Statistical Analysis

The alpha diversity was used to measure the richness and evenness within-microbial communities using the Kruskal–Wallis test. The beta diversity was used to determine the differences in the composition structure of microbial communities among samples. Permutational multivariate analysis of variance (PERMANOVA) tested the differences in beta diversity. The linear discriminant analysis (LDA) effect size (LEfSe) was used to indicate the significantly differential abundance of bacterial genus among groups, with the LDA score >3. Colonic goblet cells, mucus layer thickness, and SCFAs contents were compared using one-way ANOVA followed by post hoc multiple comparisons and analyses using Tukey’s test to compare all experimental groups. Statistical software (SPSS Inc. version 17.0, Chicago, IL, USA) was used to perform all the statistical analyses. Statistical significance was considered at *p* < 0.05 for all tests. Data were expressed as means ± standard errors of means (S.E.M). All experiments were performed in triplicate.

## 3. Results

### 3.1. DRB Supplementation Protection against Goblet Cell Loss in AOM/DSS-Induced Colitis-Associated CRC Rats

The colonic goblet cell loss was evaluated by counting the number of goblet cells per colon crypt. Figure 2A shows the histopathology of colonic tissue sections with goblet cells in the colon crypts in each experimental group. Goblet cells are mucin-secreting glands with a cup-like shape (narrow base and wide apex) located within the simple epithelium of the gastrointestinal tract. The goblet cells showed light blue color when stained with alcian blue and periodic acid-Schiff (PAS) dye. The control, DRBL, and DRBH groups do not show goblet cell loss within the colon crypts. Conversely, the induction, induction + DRBL, and induction + DRBH groups showed a loss of goblet cells.

The number of goblet cells per crypt in each experimental group is shown in Figure 2B. The number of goblet cells per crypt in control, DRBL, and DRBH groups was 25.92 ± 0.90, 25.98 ± 0.50, and 27.00 ± 0.52, respectively. There was no significant difference between the three groups. On the other hand, the induction, induction + DRBL, and induction + DRBH groups exhibited a significant decrease in the number of goblet cells (7.57 ± 0.75, 9.08 ± 0.80, and 11.04 ± 0.43, respectively) compared with the control group (*p* < 0.05). Apart from this, the induction + DRBH group showed a significant increase compared to the induction group (*p* < 0.05). These results indicated that DRB might restore goblet cell loss in the AOM/DSS-induced colitis-associated CRC rats.

### 3.2. DRB Supplementation Restored Intestinal Barrier in AOM/DSS-Induced Colitis-Associated CRC Rats

Figure 3A shows the histopathology of colonic tissue sections with mucus layers in each experimental group. The mucus secreted by goblet cells covers the surface of the epithelium of the gastrointestinal tract to lubricate the luminal contents and work as a physical barrier to bacteria and other antigenic substances present in the lumen. The mucus layer stained with alcian blue showed light blue color. The control, DRBL, and DRBH groups showed a firm mucus layer, whereas the induction, induction + DRBL, and induction + DRBH groups showed a thin mucus layer.

The mucus layer thickness in each experimental group is shown in Figure 3B. The mucus layer thickness in control, DRBL, and DRBH groups was 32.79 ± 1.69, 33.76 ± 1.75, and 34.71 ± 0.74, respectively. At the same time, thickness in DRBL and DRBH groups was slightly higher than that of the control group but showed no significant difference. In contrast, the induction, induction + DRBL, and induction + DRBH groups showed a significant decrease in the thickness (16.82 ± 0.67, 20.67 ± 0.58, and 24.30 ± 1.41, respectively) compared to the control group (*p* < 0.05). However, the induction + DRBH group showed a significant increase in the thickness compared to the induction group (*p* < 0.05). These results implicated that DRB might restore mucus layer thickness in the AOM/DSS-induced colitis-associated CRC rats related to goblet cell formation.

### 3.3. DRB Supplementation Modulated the Composition of Gut Microbiota in AOM/DSS-Induced Colitis-Associated CRC Rats

To investigate the effect of DRB on gut microbiota changes in AOM/DSS-induced colitis-associated CRC rats, 16s rRNA genes at V3–V4 regions amplified from colonic luminal content (feces), mucosa and tumor were sequenced by using Illumina sequencing. A total of 2,654,123 sequences from 66 samples in feces were generated after quality filtering steps, with an average of 40,213 sequences per sample (ranging from 2365 to 84,982 sequences). In mucosa and tumor, a total of 1,608,742 sequences from 50 samples (33 samples in mucosa and 17 samples in tumor) were generated after quality filtering steps, with an average of 30,937 sequences per sample (ranging from 9064 to 93,018 sequences). Then, after removing low-quality sequences at a quality score of 20, high-quality sequences were selected and clustered into 611 features in feces, and 898 features in mucosa and tumor of amplicon sequence variants (ASVs) processed with the QIIME 2 pipeline.

#### 3.3.1. Alpha Diversity

The rare fraction curves of alpha diversity measures reached a plateau, indicating that the sequencing depth was appropriate, and represented most of the community in the sample (Appendix A). The alpha diversity analysis was used to measure the richness and evenness within microbial communities using Pielou’s evenness, Faith’s phylogenetic diversity, Observed OTUs, and Shannon diversity index. Pielou’s evenness index represented the evenness within microbial communities. In contrast, Faith’s phylogenetic diversity and Observed OTUs index represented the richness within microbial communities, and the Shannon diversity index represented evenness and richness within microbial communities. Compared to the control group, the alpha diversity analysis in feces (Figure 4) showed a significant increase in the microbial alpha diversity only in the DRBH group according to Pielou’s evenness and Shannon diversity index (*p* < 0.05). These results specified that the microbial alpha diversity in feces did not change in the AOM/DSS-induced rats. However, DRBH supplementation increased the evenness and richness of microbial diversity within communities in feces.

In addition, the alpha diversity in the tumor (Figure 5) showed a significant decrease in the induction, induction + DRBL, and induction + DRBH groups compared to the control and the induction group in mucosa according to the Pielou’s evenness and Shannon diversity index (*p* < 0.05). Contrarily, only the induction + DRBH group showed a significant decrease in the alpha diversity in tumor compared to the control group and the induction group in mucosa according to Faith’s phylogenetic diversity and Observed OTUs index (*p* < 0.05). However, the alpha diversity among experimental groups within mucosa and tumor showed no significant difference. These results implicated that the AOM/DSS-induced colitis-associated CRC decreased the microbial alpha diversity in tumors compared to the mucosa. However, DRB supplementation did not affect the microbial alpha diversity in mucosa and tumor.

#### 3.3.2. Beta Diversity

The beta diversity was used to determine the similarities and differences in the composition structure of microbial communities using the Bray–Curtis dissimilarity, Jaccard index, and weighted and un-weighted UniFac distance. The beta diversity analysis in feces (Figure 6) and in mucosa and tumor (Figure 7) showed a significant difference among the experimental groups according to the Bray–Curtis and Jaccard index. On the contrary, the weighted and un-weighted UniFac distance showed no significant difference in microbial beta diversity among the experimental groups in feces, although in colonic mucosa and tumor, it showed a significant difference for weighted and un-weighted UniFac distance. The results above demonstrated that microbial communities’ composition structure in feces, colonic mucosa, and tumor was differed among the experimental groups using the Bray–Curtis and Jaccard index analysis.

#### 3.3.3. Bacterial Taxonomic Composition

The relative abundance of bacterial taxa in colonic feces, mucosa, and tumor at the phylum, family, and genus levels were analyzed. The taxonomic composition at the phylum level in feces is shown in Figure 8A. *Firmicutes*, *Verrucomicrobia*, *Bacteroidetes*, *Actinobacteria*, *Proteobacteria*, and *Patescibacteria* were the major phyla in the six experimental groups. Among them, *Firmicutes*, *Verrucomicrobia*, and *Bacteroidetes* were the most abundant phyla. Compared to the control and induction group, the relative abundance of *Firmicutes* was increased, and *Verrucomicrobia* was decreased in DRB supplemented groups, although the level of *Bacteroidetes* in feces was higher in the DRBH group than in the control group (Appendix A). Figure 8B shows the family level taxonomic composition. A total of 14 families with a relative abundance of more than 1% in at least one group were identified from the six experimental groups (Appendix A). In contrast to the control group, the induction group decreased the relative abundance of *Lactobacillaceae* and *Ruminococcaceae*, and increased the levels of *Erysipelotrichaceae*, and *Clostridiaceae*, respectively. However, the induction + DRBL and induction + DRBH groups increased the relative abundance of *Lactobacillaceae* and *Ruminococcaceae*. They decreased the levels of *Erysipelotrichaceae* and *Clostridiaceae* compared to the induction group. Moreover, the DRBL and DRBH groups increased the quantity of *Lachnospiraceae*, *Ruminococcaceae*, and *Prevotellaceae*. Conversely, compared to the control and induction group, the DRB supplemented groups showed a decrease in the level of *Akkermansiaceae*. The taxonomic composition at the genus level in feces was presented in Figure 8C. *Akkermansia*, *Romboutsia*, *Lactobacillus*, and *Muribaculaceae* were the major bacterial genera in feces. Appendix A shows the enriched bacterial genera in the DRB groups and in the induction groups. On the other hand, DRBL and DRBH groups revealed the increased relative abundance of *Prevotellaceae UGC-001*, *Roseburia*, and *Ruminococcaceae* compared to the control group. Furthermore, the DRBH group exhibited increased levels of *Alloprevotella*, *Butyricicoccus*, and *Ruminococcus* compared to the control group. On the contrary, the induction group has a high relative abundance of *Turicibacter*, *Clostridium sensu stricto 1*, *Enterococcus*, *Escherichia–Shigella*, and *Citrobacter*, respectively. Nevertheless, the induction + DRBL and induction + DRBH groups increased the levels of these bacterial in feces.

As shown in Figure 8D, the taxonomic composition at the phylum level in colonic mucosa and tumor of the colon mainly consists of *Firmicutes*, *Bacteroidetes*, *Verrucomicrobia*, *Proteobacteria*, *Actinobacteria*, and *Patescibacteria* phyla. *Firmicutes, Bacteroidetes,* and *Verrucomicrobia* were primarily found in the mucosa, while the abundance of *Firmicutes*, *Proteobacteria*, and *Bacteroidetes* was found in tumor (Appendix A). The induction group manifested a reduced relative abundance of *Bacteroidetes* in the mucosa, while *Firmicutes* and *Bacteroidetes* were diminished in tumors compared to the control group. In addition, the relative abundance of *Proteobacteria* was highly boosted in mucosa and tumor of the induction group. However, the induction + DRBL and induction + DRBH groups enhanced the relative abundance of *Firmicutes* and *Bacteroidetes*. They reduced the *Proteobacteria* phyla in mucosa and tumor compared to the induction group. Furthermore, in the mucosa of the DRBL group, the relative abundance of *Verrucomicrobia* was increased. The taxonomic composition at the family level is shown in Figure 8E. A total of 15 families with a relative abundance of more than 1% in at least one group were identified (Appendix A). Compared with mucosa, induction group decreased the relative abundance of bacteria in the tumor, except *Lactobacillaceae* and *Caulobacteraceae*, which were found in sufficient amounts in tumor. However, the induction + DRBL and induction + DRBH groups displayed a reduced relative abundance of *Caulobacteraceae* in tumors compared to the induction group, which manifested a decrease in *Muribaculaceae*, *Ruminococcaceae*, *Lactobacillaceae*, *Prevotellaceae*, *Eggerthellaceae*, and *Burkholderiaceae* bacterial in mucosa compared to the control group. At the same time, *Clostridiaceae 1* and *Bacteroidaceae* were increased. However, the induction + DRBL and induction + DRBH groups increased the relative abundance of *Ruminococcaceae*, *Lactobacillaceae*, *Prevotellaceae*, and *Eggerthellaceae* in the mucosa and reduced the levels of *Clostridiaceae 1*. Furthermore, the relative abundance of *Akkermansiaceae* was increased in the mucosa of the DRBL group, and the level of *Prevotellaceae* was increased in the mucosa of the DRBH group compared to the control group. At the genus level, *Lactobacillus*, *Akkermansia*, *Enterococcus*, *Romboutsia*, and *Escherichia–Shigella* have a significant abundance in mucosa and tumor of the colon (Figure 8F). Appendix A shows the enriched bacterial genera in the DRB and induction groups. Compared to the control group, the DRBL, and DRBH groups increased the relative abundance of *Prevotellaceae UCG-001*, *Roseburia*, and *Ruminococcus* in the mucosa. Moreover, the DRBL group increased the levels of *Akkermansia* in mucosa compared to the control group. Induction + DRBL and induction + DRBH groups increased the relative abundance of *Alloprevotella*, *Butyricicoccus*, *Lactobacillus*, *Prevotellaceae UCG-001*, and *Ruminococcus* in mucosa. They raised the levels of *Lactobacillus* and *Prevotellaceae UCG-001* in tumors compared to the induction group. Alternatively, the relative abundance of *Enterococcus*, *Escherichia–Shigella*, *Citrobacter*, *Clostridium sensu stricto 1*, and *Mycobacterium* in the mucosa, and *Citrobacter*, *Escherichia–Shigella*, and *Mycobacterium* in the tumor were increased in the induction group compared to the control group. Nevertheless, the induction + DRBL and induction + DRBH groups reduced the levels of *Escherichia–Shigella*, *Citrobacter*, *Clostridium sensu stricto 1*, and *Mycobacterium* in mucosa and tumor.

These results demonstrated that the bacterial taxonomic profile in feces, mucosa, and tumor was distinct from one another in this model. At the same time, DRB supplementation could alter the gut microbial composition in colonic feces, mucosa, and tumor.

#### 3.3.4. Bacterial Biomarkers

The linear discriminant analysis (LDA) effect size (LEfSe) analysis with significant differences in the bacterial genus in feces among groups is shown in Figure 9A. The LEfSe analysis showed that the DRBL group significantly increased the relative abundance of *Romboutsia* with a significant difference in LDA score (>4). The DRBH group has an abundance of three taxa, including *Ruminiclostridium 9*, *Ruminococcus 1*, and *Lachnospiraceae NK4B4* group, with a significant difference in LDA score (≥3). In the induction group, the relative abundance of *Clostridium sensu stricto 1*, *Caldicoprobacter*, and *Enterococcus* was significantly increased with an LDA score of ≥3. Similarly, while induction + DRBL group showed a remarkable increase in the relative abundance of *Ruminococcaceae UCG 009* with an LDA score of ≥3.

The LEfSe analysis of bacterial genus in colonic mucosa and tumor (Figure 9B) showed that the control group significantly increased the relative abundance of 19 taxa in the mucosa. Among these, *Ruminococcus 2*, bacterial belonging to *Muribaculaceae* and *Lachnospiraceae*, and unassigned taxa from *Muribaculaceae* showed a significant difference in LDA score (≥4). The DRBL group revealed a significant increase in the relative abundance of 12 taxa. Among these, *Akkermansia*, *Romboutsia*, and *Clostridium sensu stricto 1* exhibited a significant difference in LDA score (≥4). The DRBH group showed a notable increase in the relative abundance of 13 taxa in the mucosa. *Alloprevotella*, *Ruminiclos-tridium 9*, and some uncultured genera belonging to *Muribaculaceae* and *Lachnospiracoccaceae* showed a remarkable difference in LDA score of ≥4. A significant difference in the abundance of 3 taxa in the mucosa was found in the AOM/DSS group. There were *Blautia*, *Angelakisella*, and *ASF356* from *Lachnospiraceae* with a significant difference at LDA score ≥3. In the induction + DRBL group, the relative abundance of *Turicibacter*, *Bacteroidales bacterium*, *BB60 group* unassigned taxa, and *Candidatus Saccharimonas* in the mucosa was significantly increased with an LDA score of ≥3. Similarly, induction + DRBH group showed an increase in the relative abundance of *Lachnospiraceae NK4A136* group, *Bacteroides*, and *Prevotellaceae UCG 001* in mucosa with an LDA score of ≥3. In the tumor, the induction group showed that the relative abundance of *Enterococcus*, *Escherichia–Shigella*, and *Proteus*, significantly increased with an LDA score of ≥3. In addition, the relative abundance of seven taxa of the induction + DRBH group in the tumor was significantly increased, including *Citrobacter*, *Brevundimonas*, bacterial genera belonging to *Enterobacteriaceae*, *Mycobacterium*, *Mythylobacerium*, *Streptomyces*, and *Bosea* with an LDA score of ≥3.

### 3.4. DRB Supplementation Increased Cecal Short-Chain Fatty Acids (SCFAs) Production in AOM/DSS-Induced Colitis-Associated CRC Rats

The quantity of SCFAs in cecal contents in each experimental group is shown in Figure 10. Highest amount of acetic acid was observed in cecal contents, followed by propionic acid and butyric acid. The concentration of cecal acetic acid in control, DRBL, DRBH, induction, induction + DRBL, and induction + DRBH groups was found to be 19.50 ± 0.56, 20.12 ± 0.86, 18.74 ± 0.60, 12.83 ± 0.41, 16.80 ± 0.64, and 17.16 ± 0.51 mmol/kg wet sample, respectively. The induction group showed a significant decrease in acetic acid concentration compared to the control group (*p* < 0.05). Furthermore, the induction + DRBL and induction + DRBH groups were significantly higher in the acetic acid concentration than the induction group (*p* < 0.05). Likewise, the concentration of cecal propionic acid in control, DRBL, DRBH, induction, induction + DRBL, and induction + DRBH group were noted to be 6.18 ± 0.12, 6.47 ± 0.40, 6.23 ± 0.11, 5.20 ± 0.15, 6.08 ± 0.19, and 5.96 ± 0.10 mmol/kg wet sample, respectively. The concentration of propionic acid in the induction group was significantly decreased compared to the control group (*p* < 0.05). Moreover, the propionic acid concentration in the induction + DRBL and induction + DRBH groups tend to increase compared to the induction group. Compared to the induction group, the propionic acid concentration in the induction + DRBL and induction + DRBH group showed no significant difference. For the cecal butyric acid, the concentration of control, DRBL, DRBH, induction, induction + DRBL, and induction + DRBH group were determined to be 5.85 ± 0.10, 6.00 ± 0.33, 5.75 ± 0.24, 4.89 ± 0.12, 5.68 ± 0.17, and 5.60 ± 0.28 mmol/kg wet sample, respectively. The butyric acid in the induction group was significantly reduced compared to the control group (*p* < 0.05). In the induction + DRBL g and induction + DRBH groups, the propionic acid tends to elevate compared to the induction group. However, both groups showed no significant difference in the butyric acid concentration compared to the induction group. These results demonstrated that DRB intake tends to increase the production of SCFAs in the cecal, especially acetic acid, propionic acid, and butyric acid in the AOM/DSS-induced rats.

## 4. Discussion

Our previous investigations demonstrated that DRB have a substantial amount of dietary fiber [26]. The degradation and fermentation of these dietary fibers by bacterial enzymes produce short-chain fatty acids (SCFAs), which serve as an energy source for colonocytes and maintain intestinal homeostasis [27]. It is well known that the administration of DSS induces gut microbiota dysbiosis. Thus, we hypothesize that DRB could improve the gut microbiota in AOM/DSS-induced colitis-associated CRC model. We observed the gut microbiota communities in colonic feces, mucosa, and tumor. From the beta diversity analysis, it was noticed that DRB changes the gut microbiota profiles in feces, mucosa, and tumor. Conversely, the alpha diversity results showed that DRB did not affect the gut microbial community in mucosa and tumor. However, DRB tends to increase alpha diversity in feces. Analysis of taxonomic levels showed that *Firmicutes*, *Bacteroidetes*, *Actinobacteria*, *Verrucomicrobiota*, and *Proteobacteria* were the dominant phyla in feces, mucosa, and tumor, which follows the previous studies [28], although the bacterial taxa genus in feces, mucus, and tumors were distinct from one another in this model. In the present study, we determined that DRB could regulate gut microbiota composition in AOM/DSS-induced colitis-associated CRC rat model. At the phylum level, the relative abundance of *Proteobacteria* was increased in mucosa and tumor of the induction group. At the same time, it was lessened in the induction + DRBL and induction + DRBH groups, respectively. *Proteobacteria* are intestinal pathogens (harmful bacteria) that can cause inflammation and promote inflammation bowel disease (IBD) and CRC development and progression [29]. In addition, *Proteobacteria* was found enriched in IBD and CRC patients and AOM/DSS-induced–colitis-associated CRC mice model [30,31]. Correspondingly, the genus of *Escherichia–Shigella* and *Citrobacter* belonged to the *Proteobacteria* phylum and were found in abundance in the feces, mucosa, and tumor of the induction group. Still, it was decreased in the induction + DRBL and induction + DRBH groups. *Escherichia–Shigella* is (Gram-negative bacteria), primarily found in the gut microbiota of CRC patients, might progress the tumor formation [32]. Likewise, *Citrobacter* infection could induce immune-mediated responses and inflammation on Wnt signaling, which leads to CRC development [33]. Additionally, we determined that DRB suppressed the increase in the abundance of *Turicibacter*, *Clostridium sensu stricto 1*, and *Enterococcus* in the feces and *Clostridium sensu stricto 1* in the mucosa in AOM/DSS-induced rats. At the same time, *Mycobacterium* were decreased in the mucosa and tumor. These bacteria (*Turicibacter*, *Clostridium sensu stricto 1*, *Enterococcus*, *Mycobacterium*) are well recognized to associate with CRC development through potential mechanisms including promoting chronic inflammation, DNA damage, and the production of bioactive carcinogenic metabolites, which were increased in animal models of colitis and CRC patients [34,35,36].

In addition, we established that DRB increased the relative abundance of *Alloprevotella*, *Prevotellaceae UCG-001*, *Ruminococcaceae*, *Ruminococcus*, *Butyricicoccus*, and *Roseburia* in the feces of normal rats, while the relative abundance of *Lactobacillus*, *Alloprevotella*, *Prevotellaceae UCG-001*, *Ruminococcus*, *Ruminococcaceae*, and *Butyricicoccus* were increased in AOM/DSS-induced rats with DRB supplementation. In mucosa, we observed that DRB increased *Prevotellaceae UCG-001*, *Roseburia*, *Ruminococcus* in normal rats. In contrast, the levels of *Lactobacillus*, *Alloprevotella*, *Prevotellaceae UCG-001*, and *Ruminococcus*, and *Butyricicoccus* were increased in AOM/DSS-induced rats with DRB supplementation. In tumor, the results showed that DRB raised the *Lactobacillus* and *Prevotellaceae UCG-001* levels in AOM/DSS-induced rats. These bacteria, as mentioned above, are associated with high dietary fiber consumption. Dietary fiber is degraded to monosaccharides by fiber-degrading bacteria (*Alloprevotella* and *Prevotellaceae*) [37], which enter the cytosol and then fermented into SCFAs by SCFA-producing bacteria (*Butyricicoccus*, *Ruminococcus*, *Roseburia*, *Coprococcus*, *Eubacterium*, and *Lactobacillus)* [37]. Previous reports illustrated that the main bacteria phyla responsible for the degradation of dietary fiber are *Bacteroidetes* and *Firmicutes* [38]. Soluble fiber (e.g., pectin, gums, β-glucans, inulin) can be degraded in the ileum and ascending colon. In contrast, insoluble fiber (e.g., cellulose and hemicellulose) is exclusively fermented in the distal colon [38]. Insoluble fiber intake boosts the relative abundance of *Bacteroidetes*, *Euryarchaeota*, and *Ruminococcaceae*, together with *Prevotella*, *Phascolarctobacterium*, and *Coprococcus* at the genus level. On the other hand, intake of soluble fiber results in a higher relative abundance of the phylum *Proteobacteria* and a lower abundance of *Prevotellaceae*, along with higher bacterial genera, including *Blautia*, *Solobacterium*, *Syntrophococcus*, *Weissella*, *Olsenella*, *Atopobium*, and *Succinivibrio* [38]. Therefore, these results might be due to the high dietary fiber content (especially insoluble fiber) in DRB, which may lead to boost the abundance of fiber-degrading and SCFA-producing bacteria.

*Alloprevotella* is a fiber-degrading bacterium that plays an essential role in maintaining intestinal homeostasis and is believed to promote healthy gut microbiota in the host [39]. *Prevotellaceae UCG-001* is identified as a probiotic that supports IgA secretion and butyrate production for inhibiting inflammation [40]. A substantial decrease in the abundance of *Butyricicoccus* and *Ruminococcus* (*Ruminococcaceae* family; butyrate-producing bacterium) was observed in inflammatory bowel disease (IBD) fecal microbiota, which in turn improves the clinical outcome of CRC [41]. *Roseburia* (*Lachnospiraceae* family) produces a significant amount of butyrate from dietary carbohydrate fermentation and may be necessary to control inflammatory processes in the gut [42]. *Lactobacillus*, considered a probiotic, produces anti-microbial substances for inhibiting the growth of bacterial pathogens in the intestinal lumen, thus preventing dysbiosis and the development of CRC [43]. These bacteria were considered an anti-inflammatory factor due to their metabolite products, such as SCFAs (especially butyrate), a major energy source of colonocytes and display anti-inflammatory properties [44].

Interestingly, *Akkermansia* is a mucus-degrading bacterium (phylum *Verrucomicrobia*), colonized in the mucus layer, which is associated with mucus barrier function due to its ability to degrade mucin [45]. Furthermore, *Akkermansia* is related to high phenolic compound intake [46]. Previous studies have shown that *Akkermansia* was inversely correlated with diabetes, obesity, and other diseases [47]. In contrast, other studies suggested that *Akkermansia* might promote CRC development, since *Akkermansia* can degrade mucin, damage the mucus barrier, thus leading to the bacterial invasion of epithelial cells, and stimulate immune responses that drive intestinal inflammation and CRC development [48]. Therefore, the health-beneficial and disease-promoting properties of *Akkermansia* have been further investigated.

Therefore, these results indicated that DRB inhibited the AOM/DSS-induced colitis-associated CRC by promoting the fiber-degrading and SCFAs-producing bacteria and inhibiting the production of bacterial pathogens that alleviate the progression of CRC, thus maintaining the mucus barrier in the colon. These findings are consistent with the previous studies reported by Huan et al. The results demonstrated that defatted rice bran (DFRB) as a replacement for corns increased the intestinal wall’s thickness, *Bifidobacterium* and *Lactobacillus* levels, and decreased the level of *Escherichia coli* and *Clostridium perfringens* in the small and large intestine of finishing pigs [11]. Sheflin et al. reported that the consumption of heat-stabilized rice bran (SRB) 30 g/day for 28 days in healthy adults increased the *Bifidobacterium* and *Ruminococcus* genera and branched chain fatty acids after two and four weeks of SRB consumption [22]. Another study showed that the *Bifidobacterium longum*-fermented, and the non-fermented rice bran increased the abundance of *Roseburia*, *Lachnospiraceae*, and *Clostridiales* in the cecum and colon microbiomes in mice [49]. Parker et al. demonstrated that rice bran-modified human fecal microbiota transplantation (FMT) in AOM/DSS-treated mice decreases the neoplastic lesions in the colon. Moreover, it increases the levels of *Flavonifractor* and *Oscillibacter* (correlated with colon health) and reduces the levels of *Parabacteroides distasonis* associated with increased tumor burden [50].

In addition, in our previous study, we determined that DRB has high phenolic acids, which also contribute to intestinal bacteria [26]. These compounds have the property to promote the growth of *Bidifobacteria*, such as *Faecalibacterium prausnitzii*, *Lactobacillus* sp., and *Akkermansia muciniphila*, respectively [46]. The findings suggested that hydrolyzed bound phenolics (HBP) from rice bran supplementation improved the gut microbiota dysbiosis in high-fat diet-induced mice, which increased the relative abundance of *Bacteroides*, *Rikenellaceae*, *Allobaculum*, *Faecalibaculum*, and decreased the relative abundance of *Alistipes*, *Odoribacter*, *Butyricimonas*, *Parabacteroides*, *Romboutsia*, *Ruminiclostridium 9*, *Lachnospiraceae*, and *Erysipelotrichaceae*, respectively [51].

The significant findings highlighted that DRB promoted the enrichment of fiber-degrading bacteria, SCFA-producing bacteria, and subsequent production of SCFAs. The Bacteroidetes phylum mainly produces acetate (*Akkermansia*, *Bacteroides*, *Prevotella*, *Bifidobacterium*, *Ruminococcus*, *Clostridium*, *Streptococcus*, *Blautia*, *Coprococcus*) and propionate (*Dialister*, *Bacteroides*, *Coprococcus*, *Roseburia*, *Veillonella*, *Anaerostipes*). In contrast, the Firmicutes phylum (*Roseburia*, *Eubacterium*, *Coprococcus*, *Ruminococcus*, *Clostridium*, *Anaerostipes*) produces butyrate [19]. Herein, we measured the SCFAs, including acetate, propionate, and butyrate in the cecum, a major fermentation site in the rat. The results showed that the concentration of these SCFAs was reduced in the induction group compared to the control group. Moreover, SCFAs were higher in the induction + DRBL and induction + DRBH groups than in the induction group. These results suggested that DRB elevated the production of acetate, propionate, and butyrate in the cecum in AOM/DSS-induced rat model. The increased SCFAs in the study might be due to dietary fiber, a significant component in DRB. The increased production of SCFAs in the cecum of the induction + DRBL and induction + DRBH groups was correlated with the increase in the SCFAs-producing bacteria, including *Ruminococcus*, *Butyricicoccus*, and *Roseburia*, respectively. Furthermore, the increased SCFA production in the induction + DRBL and induction + DRBH groups was related to the decrease in pro-inflammatory markers, including TNF-α, IL-6, NF-κB, and COX-2, together with the reduction in the number of aberrant crypt foci (ACFs), and tumor formation in the colon in our previous study [26]. This is in line with the previous study that showed enzyme-treated rice fiber increased SCFAs (acetate, propionate, and butyrate) contents in the cecum and reduced inflammatory cytokines (TNF-α, IL-4, IFN-γ, IL-1β, IL-6, and IL-12p70) in serum and mucosal in DSS-induced rat [52]. Another study showed that fermented rice bran decreased the pro-inflammatory cytokine transcript levels (TNF-α, IL-1β, IL-6, and IL-17) and inflammatory cell infiltration in the colon tissue. In addition, it elevated the SCFAs production in the feces and colon and Mucin-2 (Muc2) mRNA levels in the colon in colitis mice [53].

Additionally, we investigated the effects of DRB on goblet cell loss and mucus layer thickness in the colon tissue. Previous research illustrated that the administration of DSS decreased the mucus layer thickness and increased mucus permeability in the colon, resulting in easy bacterial penetration in the inner mucus layer, which reaches the epithelial cells [54]. Bacterial invasion activates immune cell infiltration and accelerates colitis and colon cancer development. Chronic inflammation in the colon displays epithelial erosions, goblet cell depletion, crypt architectural distortion (such as shortening and loss), and mucosal fibrosis [55]. Goblet cells secrete mucin to cover the mucosal surfaces with a mucus layer lining that separates the intestinal epithelium from the lumen cavity [56,57,58]. In colonic inflammation, the epithelial cell alteration and goblet cell differentiation result in goblet cell depletion and Muc2 synthesis reductions. Correspondingly, our results showed that goblet cell depletion was found in the induction, induction + DRBL, and induction + DRBH groups compared to the control group. However, the induction + DRBL and induction + DRBH groups reduced goblet cell loss compared to the induction group. Therefore, DRB supplementation might prevent the goblet cell loss in AOM/DSS-induced rat model.

As mentioned above, the colonic mucus layer relates to goblet cells. The colonic mucus is composed of mucins (especially mucin 2) that are glycoproteins with specific O-linked glycans (O-glycans) produced by goblet cells [59,60]. The mucus layer is a part of the innate mucosal barrier, acting as the first line of immunological defense against mechanical, chemical, and pathogenic microorganism attacks and contributing to maintaining intestinal homeostasis [61]. Moreover, the mucus can lubricate the epithelial surface and cover the fecal pellet to separate bacteria from epithelium and feces. The mucus layer provides nutrients and attachment sites for bacteria [61]. Several studies suggested colonic inflammation is associated with mucus layer disruption, goblet cell depletion, and reduced Muc2 synthesis [58]. Similarly, our results showed that the mucus layer thickness was reduced in the induction, induction + DRBL, and induction + DRBH groups compared to the control group. However, the induction + DRBL and induction + DRBH groups increased the mucus layer thickness compared to the induction group. Therefore, DRB supplementation might improve the mucus layer disruption in AOM/DSS-induced rat model. This might be due to the DRB, which is a high dietary fiber. Dietary fiber is made of indigestible polysaccharides, and digestive enzymes cannot break them. Thus, dietary fiber is fermented and degraded by bacteria in the colon to produce the metabolite products such as SCFAs that enter the colonic epithelial cells (colonocytes) for their use as energy. SCFAs are oxidized through the β-oxidation pathway to generate carbon dioxide (CO_2_), which could be converted into bicarbonate (HCO_3_^−^) by carbonic anhydrase. This process promotes the stratification of the mucus layers, such as the unfolding of mucin and the resultant inner mucus layer converse to the outer layer and mucin to form a net-like structure [62]. Thus, the thickness of the mucus layer was correlated with the SCFA. There was evidence to confirm that dietary fiber is involved in increasing the thickness of the mucus layer. Desai et al. demonstrated that in mice fed a fiber-free diet, the colonic mucus layer thickness decreased and mucus layer susceptibility to bacterial pathogens increased [63]. Huawei et al. reported that the supplementation with soluble dietary fiber in a murine model of sepsis established by cecal ligation and puncture (CLP) significantly increased the mucus layer thickness and Muc2 expression in colon tissue [64]. Another study by Iain et al. found that rats fed with a fiber deficiency diet decreased the mucus layer thickness and reduced total mucus secretion over 6 h. In comparison, in rats fed with different soluble and insoluble fiber types in their diet, the mucus layer thickness and total mucus secretion over 6 h increased [65].

## 5. Conclusions

Our study showed that the AOM/DSS-induced colitis-associated CRC model altered gut microbiota composition in colonic feces, mucosa, and tumors and also reduced cecal SCFAs production, number of goblet cells, and thickness of mucus layer, while DRB supplementation modulated the gut microbiota dysbiosis by promoting the enrichment of healthy bacteria, including fiber-degrading bacteria and SCFA-producing bacteria. Subsequently, it helps reduce the abundance of harmful bacteria and stimulates the production of SCFAs. Furthermore, DRB supplementation restored goblet cell loss and improved mucus layer thickness. These findings suggested that DRB could be used as a prebiotic supplement to modulate gut microbiota dysbiosis, which decreases the risks of CRC, therefore encouraging further research into the utilization of DRB in various nutritional products to promote the production of health-beneficial bacteria in the colon. The studies and utilization of DRB as health product could assist to decrease the risk of some non-communicable diseases; in addition, it could also be useful for both the agriculture and industry sectors.

## Figures and Tables

**Figure 1 nutrients-15-01528-f001:**
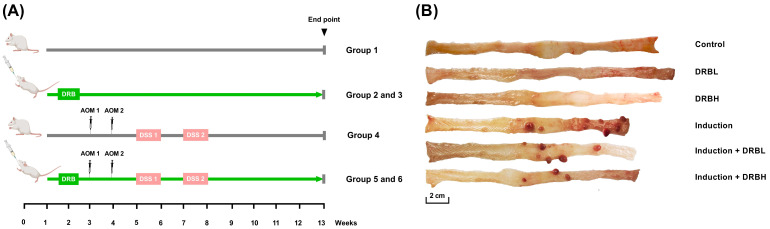
(**A**) Experimental animal design of AOM/DSS-induced colitis-associated CRC model. (**B**) Representative images of the colonic tissue in each experimental group. Rats induced with AOM/DSS appeared tumors in the colon. Bar represents 2 cm. DRBL, defatted rice bran 3 g; DRBH, defatted rice bran 6 g.

**Figure 2 nutrients-15-01528-f002:**
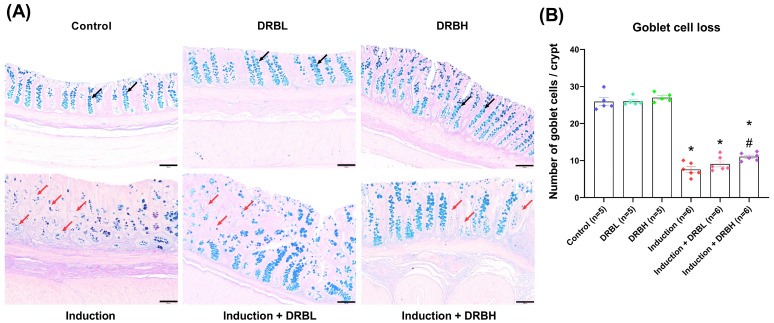
(**A**) Representative images of the goblet cells of colonic tissue sections stained with alcian blue and Periodic acid–Schiff under a light microscope at 100× magnification. The black arrow represents goblet cells which are stained with alcian blue. The red arrow is a colon crypt with goblet cell loss. Scale bars represent 100 µm. (**B**) The number of goblet cells per crypt in each group; control, DRBL, DRBH, induction, induction + DRBL, and induction + DRBH (n = 5–6/group). Data are expressed as mean ± S.E.M. Data from all experimental groups are compared using one-way ANOVA followed by Tukey’s HSD post hoc test. Mean with an asterisk (*) and number signs (#) superscript in each bar is significantly different (*p* < 0.05) when compared to the control and induction groups, respectively. DRBL, defatted rice bran 3 g; DRBH, defatted rice bran 6 g.

**Figure 3 nutrients-15-01528-f003:**
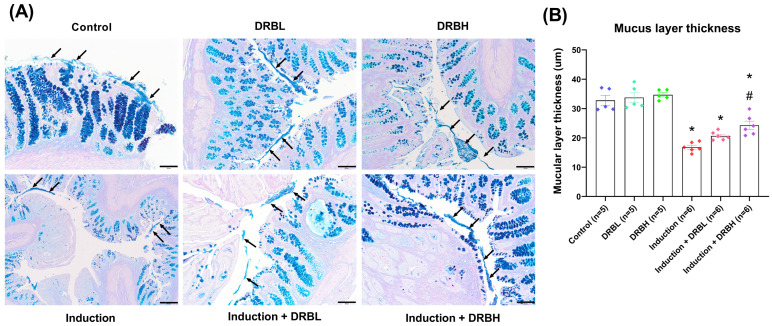
(**A**) Representative images of the mucus layer thickness of colonic tissue sections stained with alcian blue and Periodic acid–Schiff under a light microscope at 100× magnification. The black arrow represents the mucus layer. Scale bars represent 100 µm. (**B**) The muscular layer thickness (µm) in each experimental group: control, DRBL, DRBH, induction, induction + DRBL, and induction + DRBH (n = 5–6/group). Data are expressed as mean ± S.E.M. Data from all experimental groups are compared using one-way ANOVA followed by Tukey’s HSD post hoc test. Mean with an asterisk (*) and number signs (#) superscript in each bar is significantly different (*p* < 0.05) when compared to the control and induction groups, respectively. DRBL, defatted rice bran 3 g; DRBH, defatted rice bran 6 g.

**Figure 4 nutrients-15-01528-f004:**
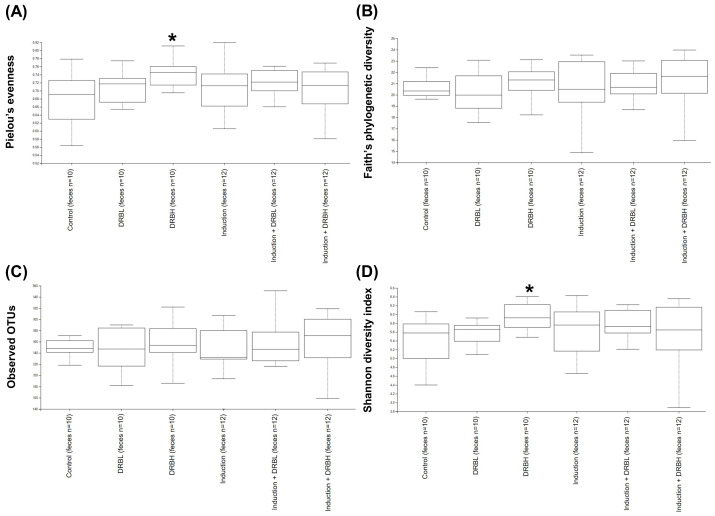
Alpha diversity analysis of gut microbiota in feces in each experimental group: control (n = 10), DRBL (n = 10), DRBH (n = 10), induction (n = 12), induction + DRBL (n = 12), and induction + DRBH (n = 12). Diversity within bacterial communities was measured by Pielou’s evenness (**A**), Faith’s phylogenetic diversity (**B**), Observed OTUs (**C**), and Shannon diversity index (**D**). Bars with asterisks (*) indicate statistically significant results for the control group (*p* < 0.05). The statistical analysis used the Kruskal–Wallis test. DRBL, defatted rice bran 3 g; DRBH, defatted rice bran 6 g; OTUs, operational taxonomic units.

**Figure 5 nutrients-15-01528-f005:**
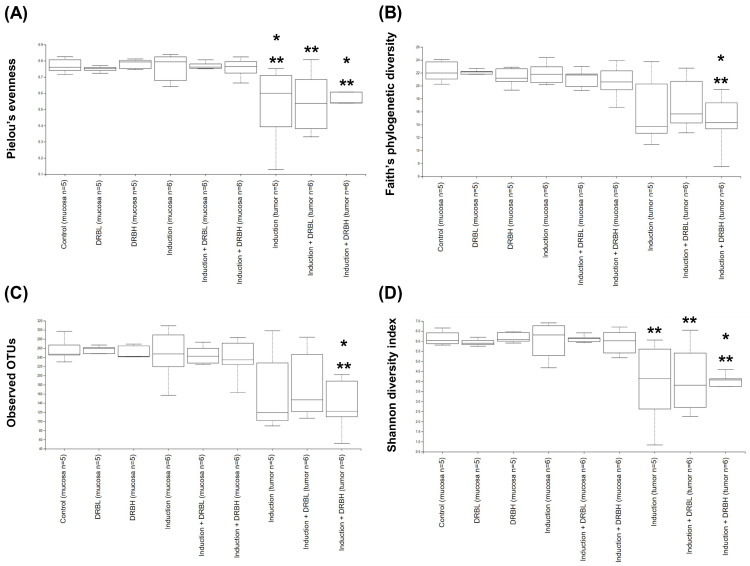
Alpha diversity analysis of gut microbiota in colonic mucosa: control (n = 5), DRBL (n = 5), DRBH (n = 5), induction (n = 6), induction + DRBL (n = 6), and induction + DRBH (n = 6) and in colonic tumor; induction (n = 5), induction + DRBL (n = 6), and induction + DRBH (n = 6). Diversity within bacterial communities was measured by Pielou’s evenness (**A**), Faith’s phylogenetic diversity (**B**), Observed OTUs (**C**), and Shannon diversity index (**D**). Bars with asterisks (*) and double asterisks (**) indicate statistically significant results for the control and induction groups in mucosa, respectively (*p* < 0.05). The statistical analysis used the Kruskal–Wallis test. DRBL, defatted rice bran 3 g; DRBH, defatted rice bran 6 g; OTUs, operational taxonomic units.

**Figure 6 nutrients-15-01528-f006:**
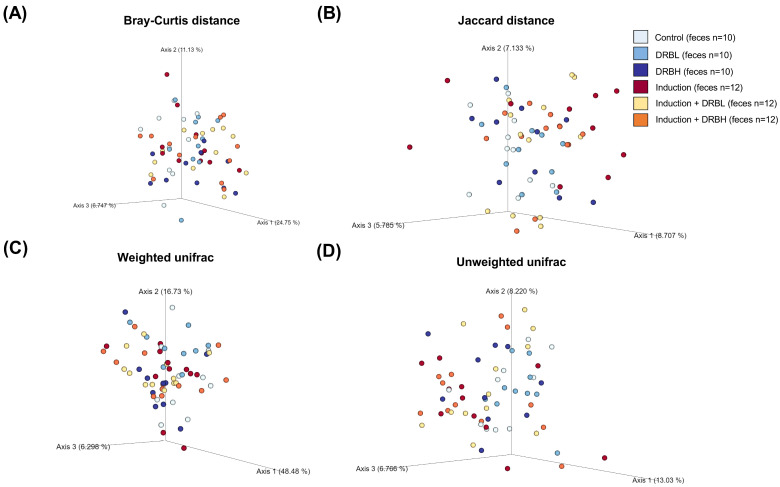
The principal coordinate analysis of beta diversity in feces based on Bray–Curtis dissimilarity (**A**), Jaccard index (**B**), and weighted (**C**) and un-weighted UniFac distance (**D**) in each experiment group: control (n = 10), DRBL (n = 10), DRBH (n = 10), induction (n = 12), induction + DRBL (n = 12), and induction + DRBH (n = 12). The differences in beta diversity were tested by Permutational multivariate analysis of variance. DRBL, defatted rice bran 3 g; DRBH, defatted rice bran 6 g.

**Figure 7 nutrients-15-01528-f007:**
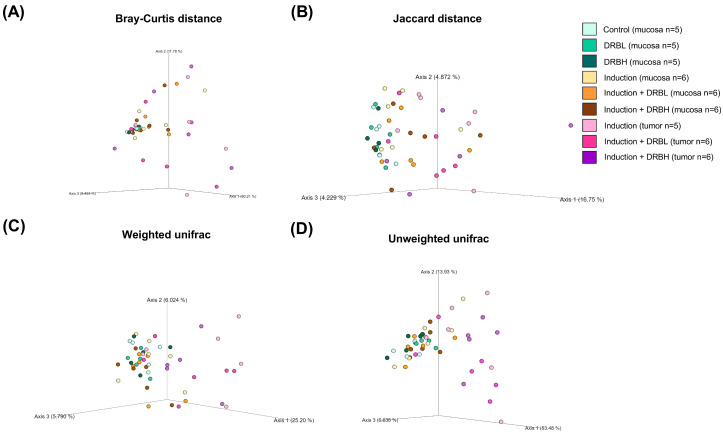
The principal coordinate analysis of beta diversity in colonic mucosa: control (n = 5), DRBL (n = 5), DRBH (n = 5), induction (n = 6), induction + DRBL (n = 6), and induction + DRBH (n = 6) and in colonic tumor; induction (n = 5), induction + DRBL (n = 6), and induction + DRBH (n = 6) based on Bray–Curtis dissimilarity (**A**), Jaccard index (**B**), and weighted (**C**) and unweighted UniFac distance (**D**) in each experiment group. The differences in beta diversity were tested by Permutational multivariate analysis of variance. DRBL, defatted rice bran 3 g; DRBH, defatted rice bran 6 g.

**Figure 8 nutrients-15-01528-f008:**
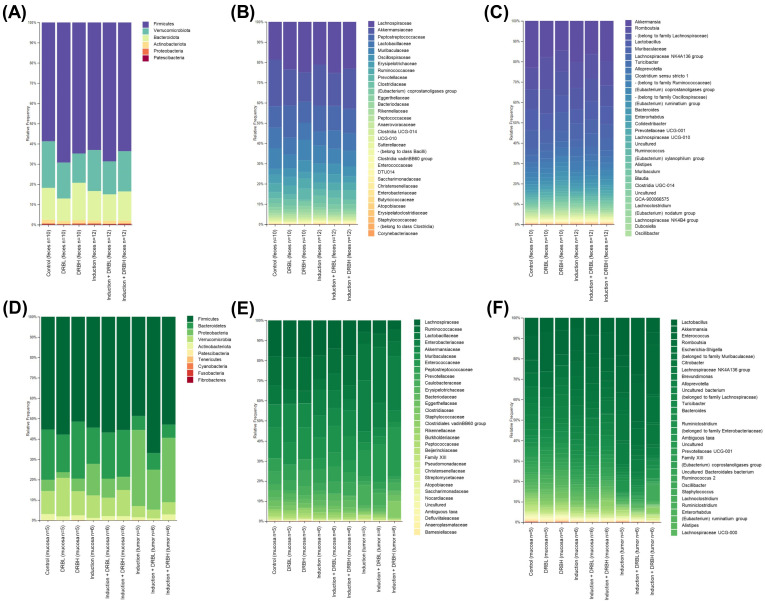
The relative abundance of bacterial taxonomic profile in feces at the phylum (**A**), family (**B**), and genus levels (**C**) in each experimental group (n = 10–12/group) and in colonic mucosa and tumor at the phylum (**D**), family (**E**), and genus levels (**F**) in each experimental group (n = 5–6/group). DRBL, defatted rice bran 3 g; DRBH, defatted rice bran 6 g.

**Figure 9 nutrients-15-01528-f009:**
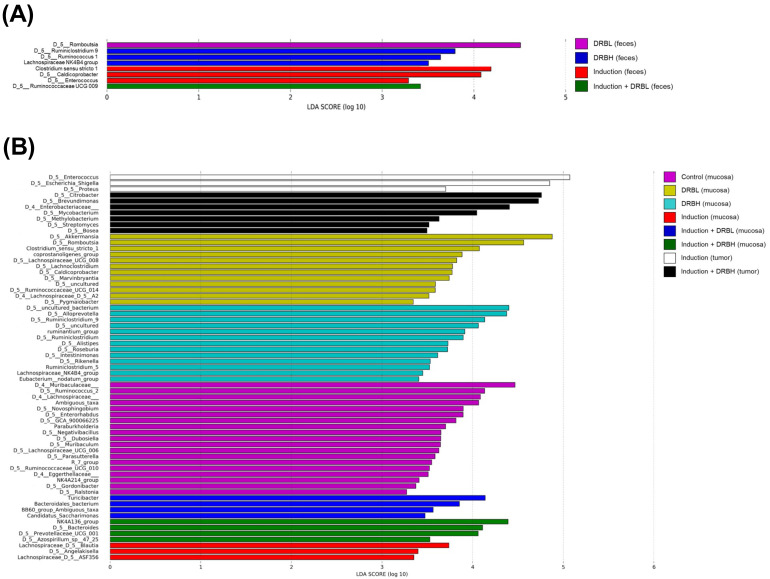
Histogram of biomarker bacterial genus in feces (n = 10–12/group) (**A**) and colonic mucosa and tumor (n = 5–6/group) (**B**) with significant difference among groups were detected by Linear discriminant analysis (LDA) effect size (LEfSe) analysis with LDA score ≥3. DRBL, defatted rice bran 3 g; DRBH, defatted rice bran 6 g.

**Figure 10 nutrients-15-01528-f010:**
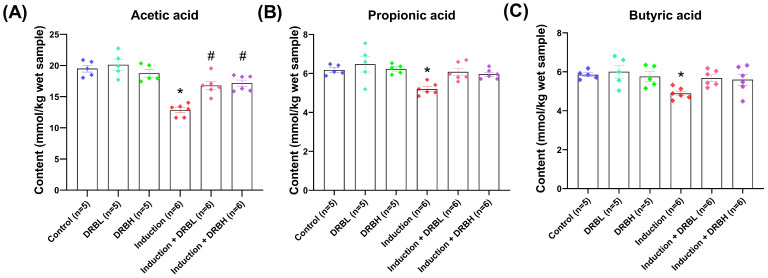
Cecal short-chain fatty acid contents (mmol/kg wet sample) in rats: (**A**) acetic acid, (**B**) propionic acid, and (**C**) butyric acid contents. Data are expressed as means ± S.E.M (n = 5–6/group). Data from all experimental groups are compared using one-way ANOVA followed by Tukey’s HSD post hoc test. Mean with an asterisk (*) and number signs (#) superscript in each bar is significantly different (*p* < 0.05) when compared to the control and induction groups, respectively. DRBL, defatted rice bran 3 g; DRBH, defatted rice bran 6 g.

## Data Availability

The datasets generated for this study are available on request from the corresponding author.

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
