# Peer review of "Role of Dietary Defatted Rice Bran in the Modulation of Gut Microbiota in AOM/DSS-Induced Colitis-Associated Colorectal Cancer Rat Model"

_nutrients, 2023, doi:10.3390/nu15061528_

Round 1

Reviewer 1 Report

1) Line 118-129: Please providing more details about experimental animals, such as the number of mice in each repeat, the initial weight with mean±SEM. Moreover, the body weight change, intestinal weight and mucosal weight are also important for readers to analysis your results.

2) Line 266-267: "50 samples (33 samples in mucosa and 17 samples in tumor)", and "Figure 3. Alpha diversity analysis of gut microbiota in colonic mucosa and tumor in each experimental group; control (n=5), DRB 3 g (n=5), DRB 6 g (n=5), AOM/DSS (n=6), AOM/DSS + DRB 3 g (n=6), and AOM/DSS + DRB 6 g (n=6). ". So where are the 17 samples in tumor?

3) The y-axis of Figure 2 and Figure 3 are too fuzzy, as well the words in Figure 4 to Figure 7, please ensure your figure quality.

4) It is suggested to displayed the Figure 8 and Figure 9 before Figure 2.

5) Figure 8, why only detected the goblet cells?

6) Figure 9, does the picture of control group is the same format with others?

Reviewer 2 Report

In the manuscript submitted for review, the authors describe the role of dietary defatted rice bran in modulating the gut microbiota in a rat model of AOM/DSS-induced colorectal cancer.

I find the topic of the manuscript extremely interesting, and "up to date", and the whole work is written well and thoughtfully. The reader's attention is undoubtedly drawn to carefully prepared figures.

My comments/questions:

1.     The Authors used male Wistar rats for the experiment. And why only male animals? what about the females? do the Authors plan to broaden their experience also with females?

2.     The Authors used CO2 to euthanize animals, but the latest research shows that decapitation is the best and least stressful way to put laboratory animals to sleep, minimizing their suffering. And in addition, CO2 has no effect on research. It is just my note for future consideration.
